# Culture of Bovine Aortic Endothelial Cells in Galactose Media Enhances Mitochondrial Plasticity and Changes Redox Sensing, Altering Nrf2 and FOXO3 Levels

**DOI:** 10.3390/antiox13070873

**Published:** 2024-07-20

**Authors:** Leticia Selinger Galant, Laura Doblado, Rafael Radi, Andreza Fabro de Bem, Maria Monsalve

**Affiliations:** 1Department of Biochemistry, Federal University of Santa Catarina, Florianópolis 88040900, Brazil; leticiagalant@hotmail.com; 2Instituto de Investigaciones Biomédicas Sols-Morreale (CSIC-UAM), Arturo Duperier 4, 28029 Madrid, Spain; lauradoblado@iib.uam.es; 3Departamento de Bioquimica y Centro de Investigaciones Biomedicas (CEINBIO), Facultad de Medicina, Universidad de la República, Montevideo 2125, Uruguay; rradi@fmed.edu.uy; 4Department of Physiological Science, Institute for Biological Sciences, University of Brasília, Brasília 70910900, Brazil; 5National Institute of Science and Technology on Neuroimmunomodulation, Rio de Janeiro 21040360, Brazil

**Keywords:** endothelial cell, glucose, galactose, mitochondria, antioxidants, oxidative metabolism, redox, FOXO3, NRF2

## Abstract

Understanding the complex biological processes of cells in culture, particularly those related to metabolism, can be biased by culture conditions, since the choice of energy substrate impacts all of the main metabolic pathways. When glucose is replaced by galactose, cells decrease their glycolytic flux, working as an in vitro model of limited nutrient availability. However, the effect of these changes on related physiological processes such as redox control is not well documented, particularly in endothelial cells, where mitochondrial oxidation is considered to be low. We evaluated the differences in mitochondrial dynamics and function in endothelial cells exposed to galactose or glucose culture medium. We observed that cells maintained in galactose-containing medium show a higher mitochondrial oxidative capacity, a more fused mitochondrial network, and higher intercellular coupling. These factors are documented to impact the cellular response to oxidative stress. Therefore, we analyzed the levels of two main redox regulators and found that bovine aortic endothelial cells (BAEC) in galactose media had higher levels of FOXO3 and lower levels of Nrf2 than those in glucose-containing media. Thus, cultures of endothelial cells in a galactose-containing medium may provide a more suitable target for the study of in vitro mitochondrial-related processes than those in glucose-containing media; the medium deeply influences redox signaling in these cells.

## 1. Introduction

The choice of energy substrates for cell culture in vitro determines cellular metabolic pathways. While high glucose levels in culture media promote cell proliferation, other monosaccharides, such as fructose and galactose, also serve as good energy sources, influencing cell metabolism. Galactose, for instance, needs to be converted to glucose through the Leloir pathway, and its oxidation to pyruvate does not generate net ATP. Therefore, cells cultured with galactose as the main carbon source rely on mitochondrial oxidative phosphorylation (OXPHOS) to produce ATP, leading to a metabolic switch where glycolysis slows down and OXPHOS is activated, unlike cells cultured in high glucose media that tend to favor glycolysis [1,2,3,4]. Importantly, the absence of glucose and serum in the medium can quickly decrease cell viability, whereas cells grown in the presence of galactose can maintain their viability over longer periods [5].

The essential role of mitochondrial OXPHOS in endothelial physiology is well established [6]. Endothelial cell mitochondria are highly coupled and operate at submaximal capacity, showing significant bioenergetic flexibility [7]. This plasticity is crucial for responding to stimuli such as blood flow, hypoxia, angiogenesis, and inflammation. Additionally, endothelial cells express PGC-1α, a key regulator of mitochondrial function, which plays a critical role in endothelial processes such as proliferation, migration, flow response, and nitric oxide production [8,9].

The formation of mitochondrial networks that undergo fusion and fission cycles in various cell types, including endothelial cells, is closely linked to mitochondrial metabolic function. Fusion is associated with “coupling efficiency” and low superoxide production [10], while fission is involved in removing damaged mitochondria, response to growth factors, and cell proliferation [11,12]. Additionally, various studies have shown that mitochondrial fission serves as an adaptive response to cellular stress and can impact the control of mitochondrial-dependent apoptosis [13]. Endothelial cells undergoing oxidative stress exhibit mitochondrial fission [14,15,16]. As the mitochondrial network is crucial for endothelial function, the choice of energy substrate for studying mitochondrial function in these cells is vital [5].

Transcription factors (TFs) that respond to the intracellular redox status, such as FOXO1/3 and Nrf2, are likely to influence metabolic processes, promoting the creation of highly energetic fused mitochondria while maintaining low levels of reactive oxygen species (ROS) and preserving cell viability, particularly in the absence of growth factors [17,18]. Currently, few studies have considered these aspects, making it challenging to reconcile cell culture findings with in vivo data on these phenomena.

Therefore, in this study, we aimed to evaluate the effects on endothelial cells’ mitochondrial dynamics and function of their exposure to a culture medium containing galactose instead of glucose. We found that these cells have greater mitochondrial oxidative respiration, enhanced mitochondrial network, and increased intercellular connectivity, which is a known important factor in the response of cells to apoptotic stimuli. In contrast, cells maintained in a glucose-containing medium display mitochondria fragmentation and decreased oxidative flux. In addition, cells maintained in a glucose-containing medium have increased Nrf2 content and undergo time-dependent variations in nuclear vs. cytosolic FOXO3 localization.

## 2. Materials and Methods

Cell culture and treatments—Bovine aortic endothelial cells (BAEC) were extracted from fresh bovine thoracic aorta as previously described by Peluffo and colleagues [9]. BAEC were cultured in growth medium (DMEM) supplemented with 10% of fetal bovine serum (FBS; Gibco/Invitrogen, Waltham, MA, USA) containing 2 mM glutamine, 100 units/mL penicillin, 100 μg/mL streptomycin, 10 mM Hepes, 25 mM glucose, and 44 mM NaHCO_3_ and incubated at 37 °C in a humidified atmosphere of 5% CO_2_. Cell suspensions were seeded in dish plates (100 × 20 mm) in 96-, 24-, or 6-well plates at different cell densities, depending on the experimental procedure. Cells at passages P4–P8 were used. When confluence reached 90%, culture media was changed to: (i) a glycolytic media, DMEM (25 mM glucose) without FBS for 3, 6, 12, 24, or 48 h, and (ii) an oxidative media, glucose-free RPMI medium with 5 mM galactose and without FBS for 3, 6, 12, 24, or 48 h. For each independent experiment, identical cells, derived from the same original culture dish, were split and then exposed to glucose and galactose media separately. Therefore, all the cells within each independent experiment had the same passage.

Oxygen consumption—Oxygen consumption rates (OCR) were measured using a Seahorse Bioscience system. To evaluate mitochondrial oxygen consumption in BAEC, cells were plated in XF24 cell culture microplates (24-well plates at a cell density of 5 × 103 cells/well) and submitted to the glycolytic (glucose) or the oxidative (galactose) protocol for 3, 12, or 48 h. A calibration cartridge (Seahorse Bioscience, Agilent Technologies Spain, Las Rozas de Madrid, Spain) was equilibrated overnight and then loaded with unbuffered cell culture media (port A), 0.6 μM oligomycin (port B), 0.3 μM FCCP (port C), and 0.1 μM rotenone plus 0.1 μM antimycin A (port D), all obtained from Sigma-Aldrich (Darmstadt, Germany). This allowed the determination of basal respiration, the maximal respiration reserve capacity, and extra mitochondrial respiration. The experiments were run with the cells exposed to the same media conditions as while in culture, but unbuffered. In all experiments, the protein concentration in each well was determined at the end of the measurements using the Pierce BCA protein assay kit (Thermo Scientific, Waltham, MA, USA) after cell lysis in RIPA buffer (Sigma-Aldrich, Darmstadt, Germany), supplemented with a protease inhibitor cocktail (Complete Mini; Roche Farma S.A., Madrid, Spain), and used to calibrate the oxygen consumption data.

MitoSOX Imaging—Mitochondrial superoxide was analyzed by labeling cells with MitoSOX Red (Molecular Probes, Carlsbad, CA, USA). BAEC were grown in coverslips in 24-well culture (1 × 105 cells/well) plates and submitted to glycolytic (glucose) or oxidative (galactose) protocols for 24 h. Then, BAEC were incubated with 3 μM MitoSOX Red for 10 min, fixed with paraformaldehyde, and analyzed by fluorescence microscopy (Leica TCS SP5, Buffalo Grove, IL, USA).

Immunofluorescence (IF)—BAEC were grown on coverslips in 24-well culture (1 × 105 cells/well) plates and submitted to the glycolytic (glucose) or the oxidative (galactose) protocol for 3, 6, 12, 24, or 48 h. At the end of the incubation period, the cells were fixed with 3.7% formaldehyde, permeabilized with 0.1% Triton, and then incubated consecutively with a primary antibody directed against Tomm22 (1:200, HPA003037, MERCK, Darmstadt, Germany) for mitochondrial dynamics analysis and Nrf2 (1:200, PA1-38312, Thermo Fisher, Waltham, MA, USA) and FOXO3 (1:100, #9467, Cell signaling) redox-related transcription factors, and then incubated with a secondary antibody (−IgG rabbit ALEXA-488 conjugate, 1:2500). Secondary antibody controls are included in Appendix A. Cells were counterstained with DAPI, mounted, and examined by confocal microscopy (Zeiss LSM 700, Obercochen, Germany), as previously described [10]. Control IFs, incubated only with the secondary antibody, were used to control for the background signal that was subtracted from the analysis.

The total α-Tomm22 IF signal (Atlas Antibodies, HPA003037, Bromma, Sweden) was used for the evaluation of the cellular mitochondrial content. Mitochondrial fission was determined as the standard deviation of Tomm22 intensity signal across the cell. Mitochondrial subcellular distribution was also evaluated using Tomm22 signal determination. The perinuclear region signal was compared to the total cytosolic signal, and the asymmetry of the signal in the perinuclear region was also evaluated by comparing the opposite max and min signals in opposite nuclear sides. The nuclei were identified by DAPI staining. This analytical procedure has already been reported [19].

Gene expression analysis—Cultured cells were washed with PBS, and total RNA was isolated using TrizolTM reagent (ThermoFisher Sci., Waltham, MA, USA) following the manufacturer’s instructions. cDNA was synthesized from total RNA preparations by reverse transcription of 1 μg of RNA using MMV reverse transcriptase (Promega Biotech Ibérica SL, Alcobendas, Madrid, Spain), in a final volume of 20 μL. The mixture was incubated at 37 °C for 45 min and then cooled for 2 min at 4 °C. The resulting cDNA was used as a template for subsequent qPCR. The primers used are listed below. Each 10 µL PCR reaction included 1 µL cDNA, 5 µL qPCRBIO SyGreen Mastermix (Cultek SL, Dutchcher Group, San Fernando de Henares, Madrid, Spain), and primers (0.3 µM). Samples were analyzed in triplicate on a Mastercycler^®^ RealPlex2 (Eppendorf Iberica SLU, San Sebastian de los Reyes, Madrid, Spain). *36B4* was used as loading control.

*Acidic ribosomal protein 36B4 (36B4)-RPLP0*-Gene ID: 286868

forward 5′-GCGACCTGGAGTCCAACTA-3′

reverse 5′-ATCTGCTGCATCTGCTTGG-3′

*Cytochrome c (Cyt c)-CYC*-Gene ID: 510767

forward 5′-GCCAATAAGAACAAAGGCATCA-3′

reverse 5′-GTTTTGTAATAAATAAGGCAGTGG-3′ 

*Mitochondrial fission protein 1 (Fis1)*-*FIS1*-Gene ID: 615565

forward 5′-GACATCCGTAAAGGCCTTGC-3′

reverse 5′-TCCATCTTTCTTCATGGCCT-3′

*Mitochondrial dynamin like GTPase (Opa1)*-*OPA1*-Gene ID: 524142

forward 5′-TGGAAAATGGTACGAGAGTCAG-3′

reverse 5′-ACTGCTGAAGGATTTCTTCC-3′ 

*Peroxiredoxin 3 (Prx3)-PRDX3* Gene ID: 281998

forward 5′-TTCGGGCTTCGCTCATCCGA-3′

reverse 5′-ATGGTATGAGGAACTGGTGCT-3′ 

*ATP synthase subunit β1 (Atp5bp)*-*ATP5BP*-Gene ID: 327675

forward 5′-TGTACCACCTCTTCCTGAACA-3′

reverse 5′-AGTGCCTGCTGTGACTTCTC-3′ 

*Superoxide dismutase 2 (Sod2)*-*SOD2*-Gene ID: 281496

forward 5′-GGAACAACAGGTCTTATCCCCCT-3′

reverse 5′-TTACTTGCTGCAAGCCGTGTATC-3′ 

*Mitofusin 2 (Mfn2)-MFN2*-Gene ID: 534574

forward 5′-TGGCGCAAGACTACAAACTG-3′

reverse 5′-TCGTCCACCAACACAGAGAG-3′

Western blotting—Whole cell extracts were prepared as previously described [9]. Protein concentration was evaluated by Lowry’s method using the RC/DC Protein Assay (Bio-Rad). A total of 18 μg of protein extract was loaded on 8/15% SDS-PAGE gels, which was then transferred to Immobilon-P membranes (Cytiva, Washington, DC, USA). Proteins of interest were identified by western blotting using the specific antibodies listed below, as described in [9]. The proteins were visualized using Clarity ECL Substrate (BioRad Spain, Alcobendas, Madrid, Spain), and the image was captured with a chemiluminescence membrane’s reader (Nirco SL, Mostoles, Madrid, Spain). The protein bands were quantified using ImageJ software. Original scanned blots are included in the Appendix A. Tubulin was used as a loading control.

α-OPA1 1:500 Sigma-Aldrich, Darmstadt, Germany Ref. HPA036926

α-Tomm22 1:2000 Atlas Antibodies, Bromma, Sweden Ref. HPA003037

α-PGC-1 α1:1000 Cayman Chemical, Ann Arbor, MI, USA Ref. 101707

α-Tubulin 1:1000 Sigma-Aldrich, Darmstadt, Germany Ref. 9026

Image analysis—ImageJ software was used for the analysis of areas, signals in an area, and cross-section signals from fluorescence and confocal microscopic images for Tomm22, Nrf2, FOXO3, and MitoSOX and from western blot bands for α-Tubulin, α-Tomm22, and α-OPA1.

Statistics—Statistical analysis and graphics were made using GraphPad PRISM^®^ software version 9 for Windows (GraphPad Software, San Diego, CA, USA). Normal (Gaussian) distribution was evaluated with the Shapiro–Wilk normality test. Significant differences among groups were evaluated by unpaired *t*-test or two-way analysis of variance (ANOVA), depending on the experimental design. Multiple comparisons were performed using Bonferroni’s post-hoc test. Results are expressed as mean ± SEM. *p* < 0.05 was considered statistically significant. *n* ≥ 4 in all experiments. The number of independent experiments for each particular experiment is indicated in the corresponding figure legend.

## 3. Results

### 3.1. Effect of Glucose- or Galactose-Containing Medium in Cellular Respiration and Mitochondrial Superoxide Radical (O_2_^•−^) Levels in BAEC

Mitochondrial oxygen consumption in BAEC conditioned in glucose or galactose medium is depicted in Figure 1. Using a Seahorse system, the mitochondrial oxygen consumption in BAEC was evaluated. BAEC are primary cell cultures, and cellular respiration can be affected by cell passage. In fact, cells with passage 4 (P4) displayed higher basal respiration rates, as well as a higher maximum respiratory reserve capacity, estimated following FCCP stimulation, than those in passages 5 (P5) or 8 (P8) (Appendix A). Therefore, it is very important to test the cellular respiration–related pathways using early cell passages to obtain robust results in primary endothelial cell cultures. In subsequent experiments with BAEC, we used P4–P8 passages, as indicated in the figure legends. In order to confirm the activation of oxidative phosphorylation in a galactose-containing medium, we compared the mitochondrial oxygen consumption rates in BAEC conditioned in galactose and glucose cell culture media (in serum deprivation conditions) at 3, 12, and 48 h (Figure 1A,B). In the absence of serum, the maximum respiratory reserve capacity after 12–48 h of incubation of BAEC conditioned in culture medium containing galactose was found to be significantly higher when compared to that observed in cells conditioned in culture medium containing glucose. Together, these data suggest that galactose increased the maximal oxidative capacity of BAEC and possibly also its metabolic plasticity.

Mitochondria are major modulators of cellular reactive species (RS). Thus, to test the impact of the media containing glucose or galactose on mitochondrial RS, we incubated BAEC with MitoSOX Red, a reagent that labels mitochondrial superoxide. We did not detect differences in mitochondrial O_2_^•−^ level when BAEC were cultivated in media containing glucose or galactose (Appendix A), suggesting that the redox balance was not significantly affected by medium manipulations.

### 3.2. Mitochondrial Dynamics of BAEC Conditioned in Glucose or Galactose Medium

Mitochondria are plastic organelles that frequently change their morphology, volume, and intracellular distribution in response to fluctuations in metabolic demands. Thus, the disruption of this balance results in mitochondrial dysfunction. Tomm22 is part of a protein translocase complex found in the outer mitochondrial membrane and which is commonly used as a marker of mitochondrial volume, morphology, and intracellular movement by immunofluorescence. As shown in Figure 2A,B, after 3, 6, 12, 24, or 48 h in a medium containing glucose or galactose in the absence of FBS, no change in mitochondrial total volume content was observed, suggesting that the observed increase in oxidative capacity in galactose media was not related to increases in mitochondrial mass. This observation was supported by western blot analysis of Tomm22 cellular content at 24 h; a similar result was also obtained for another mitochondrial protein, OPA1, a marker of cristae density, and the ratio of the two was also preserved in the two media (Appendix A). Furthermore, we did not detect significant gene expression changes in genes coding for mitochondrial proteins, including components of the electron transport chain/ETC (*Cyt c, Atp5bp*), antioxidants (*Sod2, Prx3*), and regulators of mitochondrial dynamics (*Mnf2*, *Opa1, Fis1*) at 24 h, although a general tendency to higher values could be observed for cells in galactose (Appendix A).

Nevertheless, we observed that the mitochondria of BAEC in galactose were significantly more fused than those in glucose at 12 and 24 h (Figure 2C). It was also noted that the status of mitochondria fission in BAEC exposed to galactose was stable over time (Figure 2C), suggesting that serum deprivation induced mitochondrial fission only when cells were in glucose media. Changes in the intracellular distribution of mitochondria were also evaluated, since mitochondrial function is also dependent on its subcellular localization by the determination of perinuclear/total and left/right nuclear ratios, but no significant differences were identified in the cellular response to culture medium or time of conditioning (Appendix A).

Next, we evaluated the intercellular coordination of the mitochondrial network, since intercellular contacts provide metabolic coupling that allows coordinated cellular responses within tissues that are of relevance in the response to stimuli (i.e., oxidative stress, inflammation, and apoptosis). Therefore, we decided to evaluate the intercellular variability (standard deviation) of the immunofluorescence signal for Tomm22 as a surrogate marker for intercellular metabolic coupling. We observed that cells maintained in the galactose medium displayed significantly lower intercellular variability for Tomm22 signal than cells in the glucose medium (Figure 2D,E and Appendix A). These results could be indicative of a more stable/extended intercellular contact surface that would result in a more coordinated intercellular response in galactose-treated BAEC than in BAEC maintained in a glucose medium.

### 3.3. Effect of Glucose- or Galactose-Containing Medium on Translocation and Expression of Nrf2

In order to evaluate how substrate utilization impacted on the endothelial cell capacity to respond to oxidative stress stimuli, we evaluated the basal levels and subcellular localization of the main redox-sensitive transcription factors, Nrf2 and FOXO3, that translocate to the nuclei following oxidative stimulation. We first used immunofluorescence (IF) to investigate Nrf2 levels and subcellular localization (Figure 3A). Evaluating the signal intensity of cytosolic (Figure 3B) and nuclear (Figure 3C) Nrf2, we observed a significantly higher total level of Nrf2 in cells maintained in a medium containing glucose than in cells maintained in a medium containing galactose at all times tested, and this difference was observed both in the cytosol and the nucleus. Nevertheless, when the Nrf2 nuclear/cytosolic ratio was evaluated, it was noted that at 48 h BAEC in galactose had a higher ratio than BAEC in glucose (Appendix A). These results suggest that to maintain ROS homeostasis, BAEC in glucose need higher levels of Nrf2, both in the nucleus and in the cytosol, than BAEC maintained in a galactose medium.

### 3.4. Effect of Glucose- or Galactose-Containing Medium on Translocation and Expression of FOXO3 Transcription Factor

Next, we evaluated by IF the signal intensity of cytosolic (Figure 4B) and nuclear (Figure 4C) FOXO3, as shown in Figure 4. In contrast with the Nrf2 data, at 24 h and 48 h of incubation, we observed a higher total level of cytosolic FOXO3 in BAEC maintained in a galactose medium. Consistently, at 48 h of incubation, the levels of nuclear FOXO3 were significantly higher in BAEC maintained a galactose-containing medium. Furthermore, it was noted that BAEC in glucose showed a significant time-dependent reduction in FOXO3. However, no significant difference in the nuclear/cytosol ratio was observed between in BAEC in glucose and galactose media. (Appendix A). These results suggest that galactose-treated cells may be more dependent on FOXO3 to maintain redox homeostasis than glucose-treated cells.

## 4. Discussion

The results presented in this study show that primary endothelial cells effectively modulate their metabolism in response to cell culture conditions, which significantly impacts on the mechanisms that regulate cellular redox homeostasis. Therefore, the use of culture media containing galactose instead of glucose is pertinent for the study of endothelial physiological processes involving mitochondrial function and/or redox control, such as angiogenesis, NO production, response to blood flow, interaction with immune cells, intercellular communication, and permeability. Additionally, we noted that cells in galactose exhibited higher oxidative capacity, more fused mitochondria, and seem to be more metabolically coupled through intercellular communication than BAEC in glucose media. Furthermore, we observed that cells in glucose have higher levels of Nrf2, while those in galactose have higher levels of FOXO3, suggesting that the mechanisms involved in RS control are significantly altered by culture media conditions. 

While there is widespread evidence showcasing the functional relevance of mitochondria in endothelial physiology and redox homeostasis, this aspect is often overlooked [20]. Due to the so-called Crabtree effect, well documented in the literature for different cell types, BAEC in glucose media under serum deprivation conditions exhibited lower maximal OCRs than cells maintained in galactose-containing media, suggesting an enhanced metabolic plasticity. Similarly, increased oxidative metabolism and decreased anaerobic metabolism can be observed in muscle cells incubated with galactose instead of glucose [1,21]. Cells grown in a galactose-containing medium can double their rate of oxygen consumption and, consequently, enhance their susceptibility to mitochondrial toxins compared to cells grown in a glucose-containing medium [2]. Similar observations have been found when cells are grown in low-glucose media, and this concept is currently being investigated to study the susceptibility of cancer cells to mitochondrial toxins, such as metformin, and its therapeutic applications [22].

We assessed the metabolic plasticity of BAEC in vitro and found that those cultured in a galactose-containing medium exhibited greater oxidative capacity and plasticity compared to cells in a glucose-containing medium. Although there was no change in mitochondrial volume over time or with different energy substrates, BAEC in galactose-containing media showed a net increase in oxidative capacity per mitochondrion. This finding is further supported by the observation that BAEC in glucose-containing media displayed more mitochondrial fragmentation (fission), while those in galactose-containing media maintained a higher mitochondrial fusion index. This suggests that changes in respiratory capacity were linked to the fission/fusion state and, possibly, a higher ETC content per mitochondrion rather than effects mediated by mitochondrial biogenesis or mitophagy.

Previous studies have associated increased fragmentation and decreased mitochondrial fusion with reduced mitochondria membrane potential and increased superoxide production, possibly due to reduced supercomplex formation, potentially leading to apoptosis, as noted in endothelial cells maintained in high-glucose media [23,24,25]. However, under basal conditions, BAEC treated with glucose and galactose showed similar levels of superoxide, suggesting the maintenance of redox balance in both conditions. Our data suggest that BAEC have the capacity to rapidly adjust to changing energy demands through changes in mitochondrial dynamics and, possibly, ETC content, indicating a metabolic plasticity modulated by simple changes in culture media. Furthermore, when deprived of FBS, cells maintained in glucose-containing media can quickly induce apoptosis, while cells maintained in galactose-containing media are more resistant to growth factor deprivation, making the former conditions more suitable for testing responses requiring growth factor deprivation. Additionally, analysis of intercellular mitochondrial variability revealed that cells in glucose-containing media exhibited reduced intercellular communication, potentially due to the smaller cell–cell contact zone affecting data variability. Conversely, in galactose-containing media, intercellular metabolic coupling appeared greater, suggesting an enhanced ability to coordinate responses to external stimuli such as oxidative stress. Importantly, previous studies have linked mitochondrial activity in endothelial cells to improved and more stable cellular contacts, facilitating the formation of stable vessel structures and coordination during cell migration [8,9].

There is a well-documented interplay between metabolic and redox regulatory pathways. Thus, we investigated the impact of cell culture on the two main redox transcriptional regulators in BAEC: Nrf2 and FOXO1/3. Nrf2 is known to activate cellular antioxidant responses, promoting the synthesis of crucial enzymes during oxidative processes [26,27]. We observed that the nuclear and cytosolic content of Nrf2 in endothelial cells maintained in a glucose-containing medium was higher compared to those in a galactose medium, suggesting that Nrf2 may play a predominant role in the control of redox homeostasis in BAEC cultured in glucose conditions [28]. To the best of our knowledge, this study represents the first investigation into the involvement of Nrf2 in endothelial cells in a glycolytic environment in the absence of FBS.

FOXO3′s role in redox control is intricately linked to metabolism and vascular cell physiology [29]. It has been established that endothelial cells cultured in high-glucose media (standard medium containing approximately 25 mM) exhibited elevated RS generation, Akt inhibition, and therefore high FOXO3 activation, correlating with cellular apoptosis [30]. In tumor cells, which exhibit highly active glycolytic metabolism, elevated nuclear levels of FOXO3 were correlated with apoptosis and reduced cell survival [31,32,33]. However, FOXO3a activity can also promote cell survival when activated under conditions favoring oxidative phosphorylation, especially through the induction of antioxidant genes [11]. Our findings indicated that after 12 h of incubation, BAEC maintained in glucose-containing media exhibited reduced total FOXO3 content, whereas those in galactose-containing media maintained constant FOXO3 levels. This suggests that FOXO3 levels are better preserved under galactose conditions and underscores the potentially predominant role of FOXO3 in redox control. 

Endothelial cells are particularly vulnerable to oxidative stress due to their direct exposure to the vascular environment and frequent contact with RS. If not detoxified, RS can lead to endothelial dysfunction. Therefore, the modulation of transcription factors related to antioxidant defenses, such as Nrf2 and FOXO3, is crucial for preserving endothelial integrity. However, it is essential to exercise caution when studying the activation of these transcription factors and the mitochondrial function, considering the experimental protocol used. Experimental protocols that maintain physiological levels of transcription factors such as Nrf2 and FOXO3, as well as preserving mitochondrial modulation and function, are essential for accurately elucidating the mechanism of action of certain drugs. Depending on the energy substrate used, the mechanism of action of these molecules can be altered, potentially leading to erroneous conclusions.

In this study, we conclude that experimental protocols using primary endothelial cells maintained at standard glucose concentrations and without FBS exhibited increased mitochondrial fission and reduced respiratory reserve capacity, indicative of glycolytic metabolism. These conditions yield greater intercellular variability, likely stemming from a smaller mitochondrial cell network, potentially compromising cell–cell contact and communication. Conversely, BAEC cultured in a galactose-containing medium exhibited enhanced connectivity within the mitochondrial network and increased respiratory reserve capacity. Larger mitochondrial networks may augment cell–cell contact surface, thereby potentially reducing intercellular variability.

The significance of these alterations on redox control is underscored by the observation that BAEC cultured in media with standard glucose concentrations exhibited elevated levels of Nrf2 and decreased levels of FOXO3 compared to BAEC incubated in galactose (see Figure 5). This is consistent with prior findings and elucidates the relationship between oxidative metabolism and redox regulatory networks.

## 5. Study Limitations

This study was conducted solely on a single cell type and species; thus, the generalizability of the findings to other cell types and species remains to be established.The analytical procedure employed for assessing mitochondrial dynamics does not include high-resolution TEM analysis, and the derived conclusions would require further validation, including the analysis of proteins involved in the regulation of mitochondrial dynamics.Since we did not assess the antibodies used in immunofluorescence on knock-out cells, some signals may stem from non-specific staining of the samples.Further testing would be required to determine if the observed nuclear localization changes do actually impact on Nrf2 and FOXO3A transcriptional activity.

## Figures and Tables

**Figure 1 antioxidants-13-00873-f001:**
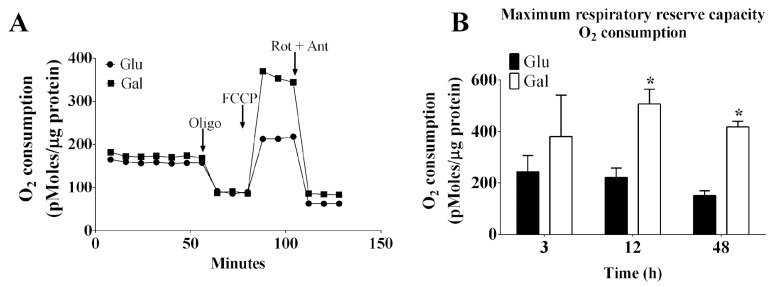
Effect of glucose- or galactose-containing medium (without FBS) on mitochondrial respiration and mitochondrial O_2_^−^ production in BAEC (P4–P6). (**A**) Representative respirometry assay of BAEC conditioned in culture media containing glucose (Glu) or galactose (Gal) for 12 h. Arrows indicate the time of adding the oligomycin, FCCP, and rotenone plus antimycin. (**B**) Maximum respiratory reserve capacity: O_2_ consumption after the addition of FCCP. Each independent experiment was performed using cells in the same cell passage exposed to glucose or galactose media. O_2_ consumption rates were corrected for protein concentration. Data were represented as mean ± SEM (n = 4). * *p* < 0.05 indicates statistical difference between the groups (glucose vs. galactose), by two-way ANOVA followed by Bonferroni’s post-hoc test. Data were represented as mean ± SEM (n = 4). * *p* < 0.05 indicates statistical difference between the groups (glucose vs. galactose), by unpaired *t*-test.

**Figure 2 antioxidants-13-00873-f002:**
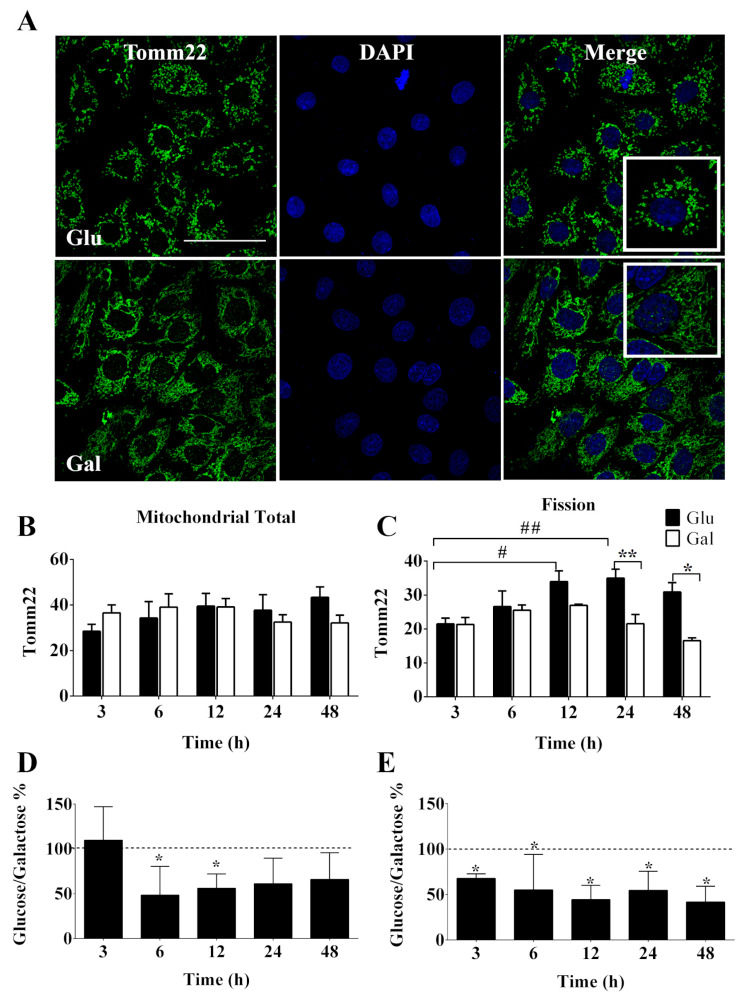
Effect of glucose- or galactose-containing medium on mitochondrial dynamic and intercellular variability in BAEC (P4-6). (**A**) Representative image of immunofluorescence from Tomm22 (green) and DAPI (blue) in BAEC for 24 h; the white bars represent 50 μm. (**B**) Quantification of mitochondrial total for (**C**) Mitochondrial fission. Intercellular variability was performed by standard deviation for (**D**) mitochondrial total and (**E**) mitochondrial fission. Each independent experiment was performed using cells in the same cell passage exposed to glucose or galactose media. Data in graphs represent mean ± SEM (n = 4). * *p* < 0.05, ** *p* < 0.01 indicates a statistical difference between the groups of glucose (Glu or dashed line) and galactose (Gal or black bars), # *p* < 0.05, ## *p* < 0.01 indicates time-dependent statistical difference between the responses to Glu and Gal by two-way ANOVA, followed by Bonferroni’s post-hoc test.

**Figure 3 antioxidants-13-00873-f003:**
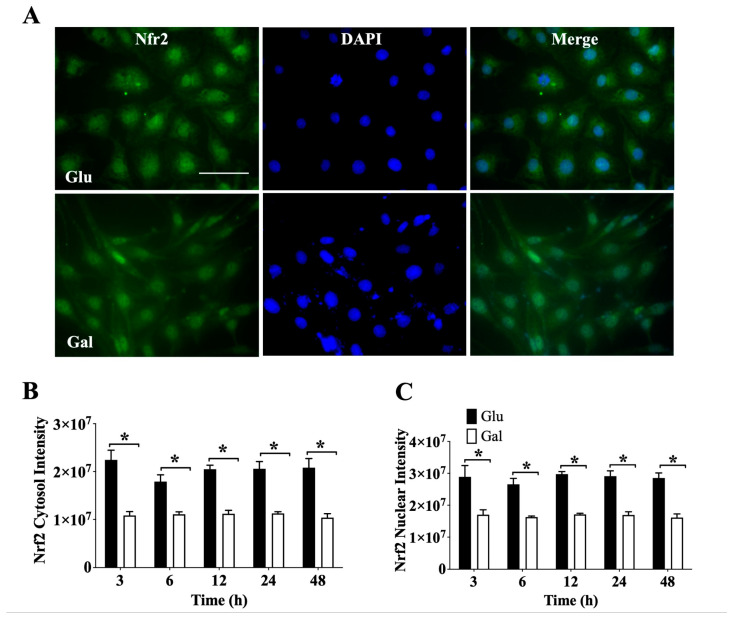
Effect of glucose- or galactose-containing medium on Nrf2 expression in BAEC (P4-6). (**A**) Representative image of immunofluorescence from Nrf2 (green) and DAPI (blue) in BAEC after 24 h of incubation with glucose or galactose; white bars represent 50 μm. (**B**) Intensity of cytosolic Nrf2, (**C**) intensity of nuclear Nrf2. Each independent experiment was performed using cells in the same cell passage exposed to glucose or galactose media. Data were represented as mean ± SEM (n = 4). * *p* < 0.05 indicates statistical difference between glucose (Glu) and galactose (Gal) conditions by two-way ANOVA, followed by Bonferroni’s post-hoc test.

**Figure 4 antioxidants-13-00873-f004:**
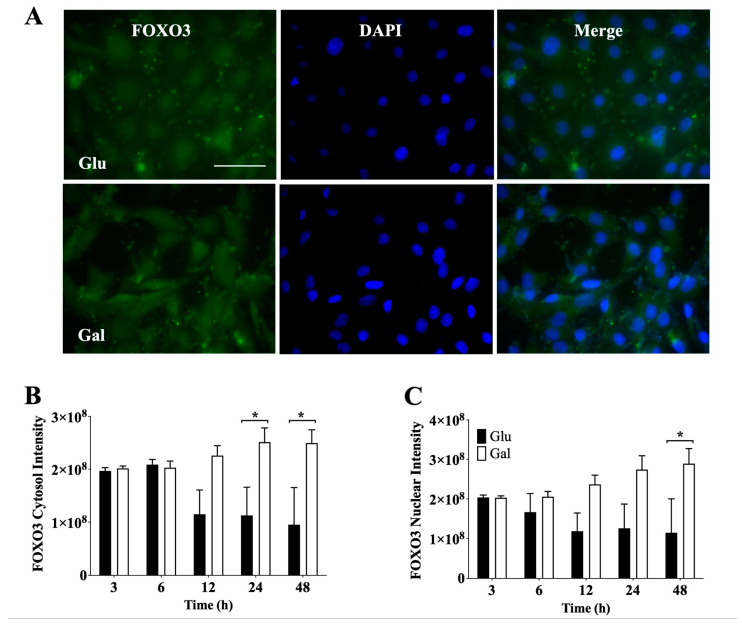
Effect of glucose- or galactose-containing medium on FOXO3 expression in BAEC (P4–6). (**A**) Representative image of IF from FOXO3 (green) and DAPI (blue) in BAEC after 24 h of incubation with glucose or galactose; white scale bars represent 50 μm. (**B**) Intensity of cytosolic FOXO3, (**C**) intensity of nuclear FOXO3. Each independent experiment was performed using cells in the same cell passage exposed to glucose or galactose media. Data were represented as mean ± SEM (n = 4). * *p* < 0.05 indicates statistical difference between glucose (Glu) and galactose (Gal) conditions by two-way ANOVA, followed by Bonferroni’s post-hoc test.

**Figure 5 antioxidants-13-00873-f005:**
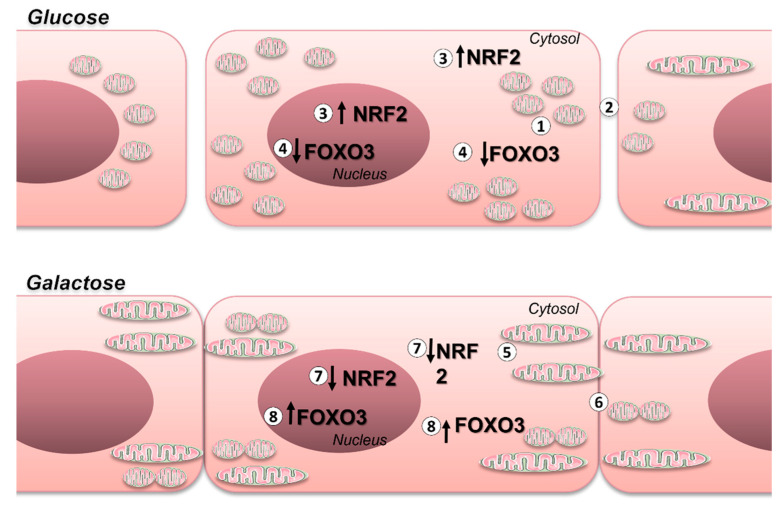
Comparison between glycolytic and oxidative metabolism in mitochondrial modulation and activation of Nrf2 and FOXO3 transcription factors in endothelial cells. Comparison of BAEC maintained in galactose- and glucose-containing media: glucose-exposed cells show greater mitochondrial fission and less reserve respiratory capacity (1), indicating activation of glycolytic metabolism, greater intercellular variability (2), which can be explained by the smaller cellular mitochondrial network. This factor can impair cell–cell contact and cell communication, lead to greater nuclear and cytosolic Nrf2 expression (3), and lower nuclear and cytosolic FOXO3 expression during prolonged incubation periods (4). In contrast, BAEC maintained in medium containing galactose showed greater connection of the mitochondrial network (mitochondrial fusion) and greater reserve respiratory capacity, possibly activating the oxidative pathway [1] (5), and greater mitochondrial networks appear to increase the cell–cell contact surface and consequently be related to the observed reduced intercellular variability (6) and lower nuclear and cytosolic Nrf2 levels (7); nuclear and cytosolic expression of FOXO3 is constant at all incubation times (8).

## Data Availability

Supporting data is included as Appendix A.

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
