# Peer review of "Culture of Bovine Aortic Endothelial Cells in Galactose Media Enhances Mitochondrial Plasticity and Changes Redox Sensing, Altering Nrf2 and FOXO3 Levels"

_antioxidants, 2024, doi:10.3390/antiox13070873_

Round 1

Reviewer 1 Report (Previous Reviewer 1)

The major experimental problem is the strong dependency (3-fold variation !!!) of glucose-dependent BAEC respiration on passage number as shown by the authors in Suppl. Fig. 2. This effect easily could generate the 2-fold differences presented in Fig. 1. To convince the critical reader the authors should provide data for BAEC respiration on glucose and galactose, respectively, at well-defined passages and show their subsequent alterations with increasing passage numbers.

1. The authors claim on line 221: 'In subsequent experiments with BAEC, we used P4-P8 passages.' That is in light of their data shown in Suppl. Fig. 2 ( 3-fold variation of oxygen consumption between passages 4 and 8 )  not precise enough. In the legend of Fig. 1 the passage number is not provided.

2. Also for all other Figures passage numbers should be provided, since there might be strong passage number-dependent expression changes.

Minor points:

3. Remove the numbering in the beginning of the text (lines: 247, 273, 296 ....)

4. Line 229: Why galactose should increase metabolic plasticity?

Author Response

Q1. The major experimental problem is the strong dependency of glucose-dependent BAEC respiration on passage number as shown by the authors in Suppl. Fig. 2. This effect easily could generate the 2-fold differences presented in Fig. 1. To convince the critical reader the authors should provide data for BAEC respiration on glucose and galactose, respectively, at well-defined passages and show their subsequent alterations with increasing passage numbers.

Answer. We have worked with primary cells for many years and are fully aware of the variations related to passage number. To control for this unavoidable but critical issue we do two things, first we tightly restrict the passages used (P4-P8), actually, most of the data presented was derived from P4-P6, and second, when conditions are compared all the cells are in the same passage, that means that for each independent experiment identical cells (derived from the same culture dish) were exposed to glucose or galactose media, In other words, cell plating for each experiment was conducted with cells at the same passage. This ensures that the observed effects are due to variations in the carbon substrate used (glucose and galactose) and not related to cell passage-dependent variation. Actually, it was from the awareness of this issue that we included in the Supplementary Figure 2, showing that OCR varies depending on the passage number.

To clarify this point, we have now included the following text in the Materials and Methods section:   “For each independent experiment identical cells, derived from the same original culture dish, were split and then exposed to glucose and galactose media separately. Therefore, the cells within each independent experiment had the same passage.”

Q2. The authors claim on line 221: 'In subsequent experiments with BAEC, we used P4-P8 passages.' That is in light of their data shown in Suppl. Fig. is not precise enough. In the legend of Fig. 1 the passage number is not provided.

Answer. As mentioned above, we were very aware of the need to minimize the effects of cell passages on all evaluated parameters. To clarify this point have now made sure that the cell passages used are referred to in all figures, and included the following the following statement in the figure legends: "Comparative analyses were performed using cells in the same cell passage.”

Q3. Also, for all other Figures passage numbers should be provided.

Answer. We have now included a reference to the passage numbers in all the Figure Legends.

Q4. Remove the numbering in the beginning of the text (lines: 247, 273, 296 ....).

Answer. Thank you for pointing out this mistake, we have now removed the numbers.

Q5. Line 229: Why galactose should increase metabolic plasticity?

Answer. When mitochondria are activated and fused, as shown here in galactose media, it has been reported that the quality control systems, that control mitochondria turnover, activation and de-activation are more active, see for example: https://doi.org/10.14814/phy2.12470

Reviewer 2 Report (Previous Reviewer 2)

The authors addressed the majority of my concerns and the manuscript has been significantly revised.

Only one or two grammar mistakes  (i.e. page 2, line 58) were found but can be easily fixed through proofreading.

Author Response

Q1. Only one or two grammar mistakes (i.e. page 2, line 58) were found but can be easily fixed through proofreading.

Answer. We have revised and corrected grammar errors, corrections are marked in blue.

Reviewer 3 Report (Previous Reviewer 3)

I thank the authors for addressing all the concerns

NA

Author Response

Q. I thank the authors for addressing all the concerns.

Answer. Thank you!

Round 2

Reviewer 1 Report (Previous Reviewer 1)

My concerns have been addressed accordingly.

My concerns have been addressed accordingly.

This manuscript is a resubmission of an earlier submission. The following is a list of the peer review reports and author responses from that submission.

Round 1

Reviewer 1 Report

The resubmitted paper has been considerably improved and attempts for better quantification of mitochondria were made. But still some problems remain.

1. The reference genes used the experiments should be always mentioned. 

2. What about the specificity of ABs used in the study? If no data are available at least a critical phrase regarding potential limitations of this study due to antibody specificity should be introduced.

1. The reference gene used in the new WB experiments (Suppl. Fig. 3A) should be mentioned.  Moreover, the error bar are pretty high. Can mitochondrial quantity differences really excluded?

2. What about the specificity of ABs used for Nrf2 and Foxo3? This regards particularly to Figs. 3 and 4.

Reviewer 2 Report

This is an in vitro study looking at the impact of galactose (vs. glucose) on endothelial cell mitochondrial dynamics and redox sensing transcriptional factor Nrf2 and FOXO3 levels. While ample data has supported the hypothesis that galactose media maintains mitochondrial plasticity and redox homeostasis under serum-free culture conditions, major concerns have been raised regarding the marginal changes presented in the dataset and the disconnection between TF and the regulation of mitochondrial morphologies. These major concerns are listed below:

1.       It is well documented that low glucose culture conditions will switch glycolysis to mitochondrial oxidative phosphorylation in ECs, thus increasing OCR and oxygen-derived reactive species. This part of work has been replicated by the study here.  The most interesting observation will be the higher mitochondrial fusion and lower mitochondrial fission by galactose culture when compared to glucose culture. However, there are multiple problems that have made the observation invalid:

A.      DMEM media was used for glucose culture, while RPMI1640 was used for galactose culture. The authors did not explain why two different basal media were used, especially since they contain different levels of glucose, calcium, and phosphate. DMEM contains higher calcium (1.8 mM) and lower phosphate (1 mM) than RPMI 1640 (0.8 mM of calcium and 5 mM of phosphate) DMEM (low glucose) has a lower concentration of glucose (1 g/L) than RPMI 1640 (2 g/L), which can be used as the medium for galactose treatment.

B.      The high glucose condition (25mM) was considered to mimic hyperglycemia, a pathological condition different from healthy condition (5mM glucose). The galactose at physiological levels were negligible (when compared with glucose) and 5mM galactose was considered to mimic galactosemia, an inherited metabolic disorder (ranging from 5-20mM).  The clinical relevance of comparing these two pathological conditions without using corresponding controls.

C.      The mitochondrial morphology staining by TOMM22 was interesting (Fig 2A), but the way the author quantified it was not accurate nor specific to mitochondrial fusion vs. fission (Fig 2B and 2C). Larger magnification should be used and the tubular mitochondria length should be quantified.

D.      The data of the Mito fusion vs. Mito fission molecules was not supportive of the hypothesis at all. There was NO difference in the protein or mRNA levels in Opa1 between glucose and galactose (Supplement Fig 3). However, the author stated that there was a difference. Drp1 was not measured.  The mitochondrial dynamic proteins are essential markers and molecules that should have been investigated in detail, but either not supportive of the hypothesis or missing in the current study.

2.       The Nrf2 nuclear enrichment appeared to be significantly higher in galactose exposure; the staining was not an accurate assay to suggest higher Nrf2 activity. An Nrf2 antioxidant response element Reporter assay is needed to indicate the transcriptional activity of Nrf2.

3.       The staining for FOXO3 showed marginal (or no) differences between glucose culture and galactose culture (Fig 3 and Fig 4) despite the significance presented in the bar graphs.  

4.       There were disconnections between Nrf2/FOXO3 and mitochondrial plasticity changes under glucose vs. galactose culture. It appeared that these molecule changes were parallel to the mitochondrial changes. 

None. 

Reviewer 3 Report

In this manuscript, Leticia Selinger Galant and colleagues describe the effect of galactose and glucose in the culture of bovine aortic endothelial cells. The authors conclude that Nrf2 and FOXO3 levels and location are responsible for mitochondrial plasticity and redox status. These results have significant implications for improving the culture of these cells. However, there are some major concerns to solve before accepting the manuscript for publication:

-              - The authors addressed measuring ROS levels by using MitoSOX only. Although there are no significant differences, I recommend using other dyes to monitor other ROS sources, not only mitochondrial superoxide.

-      - The conclusion of the study is quite risky, given the experiments shown. If Nrf2 and FOXO3 are involved in modulating ROS and mitochondrial dynamics, it should be confirmed by using activators/inhibitors of these proteins. In addition, I suggest trying to mimic glucose or galactose conditions by using drugs that alter redox signaling.  

-       -  Seahorse assays were performed by adding glucose or galactose, respectively, or just adding glucose in the Seahorse media as recommended by the manufacturer. Please indicate it in Material and Methods and discuss using galactose instead of glucose in the Seahorse assay.

-            The quality of Figures 3 and 4 is poor.